# Chiral phosphoric acid-catalyzed enantioselective phosphinylation of 3,4-dihydroisoquinolines with diarylphosphine oxides

Yongbiao Guo [1✉], Ning Li[1], Junchen Li[1], Xiaojing Bi[1], Zhenhua Gao [1✉], Ya-Nan Duan [2✉] & Junhua Xiao[1✉]

Chiral phosphorous-containing compounds are playing a more and more significant role in several different research fields. Here, we show a chiral phosphoric acid-catalyzed enantioselective phosphinylation of 3,4-dihydroisoquinolines with diarylphosphine oxides for the efficient and practical construction of a family of chiral α-amino diarylphosphine oxides with a diverse range of functional groups. The phosphine products are suitable for transforming to several kinds of chiral (thio)ureas, which might be employed as chiral ligands or catalysts with potential applications in asymmetric catalysis. Control and NMR tracking experiments show that the reaction proceeds via the tert-butyl 1-(tert-butoxy)-3,4-dihydroiso-quinoline-2(1H)-carboxylate intermediate, followed by C-P bond formation. Furthermore, computational studies elucidated that the hydrogen bonding strength between the phosphonate and iso-quinolinium determines the stereoselectivity of the phosphinylation reaction.

[1] State Key Laboratory of NBC Protection for Civilian, Beijing 102205, China. [2] Chemistry and Chemical Engineering Guangdong Laboratory, Shantou 515031, China. ✉email: van87120@126.com; gaozhenhua1223@163.com; duanyanan008_work@163.com; xiao.junhua@pku.edu.cn

C hiral phosphorous-containing compounds have attracted more and more attention due to their prevalent applications as chiral ligands and organocatalysts for asymmetric reactions[1–6], potential biological activity[7–11] and utilities in the field of material science[12–14]. Chiral α-amino diarylphosphine oxide represents one of the most significant organophosphorous compounds, which was always directly functioned as organocatalysts[15,16] or utilized as the key building blocks in chiral ligand synthesis. What's more, recent research also revealed that chiral α-amino diarylphosphine oxides can serve as surrogates for α-amino phosphonic acids and their phosphonate esters which might lead to important biological activity discoveries[17,18]. Chiral 1,2,3,4- tetrahydroisoquinoline (THIQ) is one of the most important "privileged scaffolds" present in natural products[19–22], bioactive molecules[23–26], and chiral catalysts[27–32]. Therefore, combining the substructure of α-amino diarylphosphine oxides and THIQ into a single molecule would be of great utility, no matter in discovery of new compounds with potential biological activities or development of new ligands and organocatalysts.

Catalytic enantioselective phosphinylation of imines with diarylphosphine oxides is the most straightforward approach to chiral α-amino diarylphosphine oxides (Fig. 1a). To the best of our knowledge, there are only three reports on the enantioselective synthesis of α-amino diarylphosphine oxides up to now. The first example of a heterobimetallic lanthanoid complex being employed to catalyze the asymmetric addition of diphenylphosphine oxide to cyclic imines was reported by Shibasaki et al. Under such catalytic conditions, high yields and good to excellent enantioselectivities were well achieved[33]. After this pioneering work, Antilla et al. reported the chiral magnesium BINOL phosphate-catalyzed enantioselective phosphinylation of substituted acyclic imines with diphenylphosphine to give products with up to 96% ee[34]. Shortly afterward, Gong and co-workers reported the first enantioselective phosphination of acyclic imines with diphenylphosphine oxide using chiral squaramide as a hydrogen bonding organocatalyst, giving structurally diverse α-amnio phosphine oxides with good to excellent enantioselectivities[35]. Therefore, it is highly desirable for synthetic chemists to develop new enantioselective protocols for construction of α-amino diarylphosphine oxides.

Moreover, the enantioselective protocol for constructing chiral phosphine moiety at C1 position of THIQ was rarely explored. The only work in the field was reported by Mukherjee et al. who applied chiral thiourea as the catalysts in the asymmetric synthesis of 1,2,3,4-tetrahydroisoquinoline-1-ylphosphonates (Fig. 1b)[36]. However, this method required harsh reaction conditions (−80 °C) and preformed silyl phosphite, and only four examples with moderate to good enantioselectivities. In our very recent work, we successfully realized enantioselective phosphonation of isoquinolines for construction of chiral α-aminodiarylphosphine oxide via chiral phosphoric acid-catalyzed dearomatization of isoquinolines[37]. This protocol was found to have obvious limitations owing to the difficulties lying in the further transformation of the products ascribed to its facile racemization via thermodynamically favored aromatization process. To the best of our knowledge, there is no report of asymmetric phosphinylation of 3,4-dihydroisoquinolines that utilizes diarylphosphine oxides as the nucleophiles.

It is well known that chiral phosphoric acids (CPAs) play significant roles in the research field of organocatalysis[38–44]. We are keeping interests in for its application in new asymmetric reactions and useful synthesis[45–47]. Herein, we report on the chiral phosphoric acid catalyzed asymmetric phosphinylation of 3,4-dihydroisoquinolines using secondary phosphine oxides to construct the scaffold of chiral THIQ with phosphine oxide at C1 position (Fig. 1c). This catalytic system shows quite good functional group tolerance and wide substrate scope. The applicability of this protocol was well elucidated via gram-scale synthesis and derivatization of the products with the formation of different substituted chiral (thio)ureas that could possibly function as chiral organocatalysts or ligands in asymmetric catalysis. Furthermore, the catalytic cycle was well depicted on the basis of experimental results and DFT calculations.

## Results and discussions

As our continuous research interests for developing methodology on constructing chiral phosphine-containing compounds, we explored the enantioselective phosphinylation of cyclic imines to chiral α-amino phosphine oxide. Readily available 3,4-dihydroisoquinoline (1a) and diphenylphosphine oxide (2a) were selected as the model substrates. After initial trials, it was found that the title phosphinylation reaction took place smoothly when 3,4-dihydroisoquinoline was firstly treated with Boc₂O. In order to achieve satisfying enantiomeric excess, 73 chiral phosphoric acids (CPA) were systematically investigated (see Supplementary Table 1 for details) and part of the results were summarized in Table 1. Firstly, a series of chiral phosphoric acids based on BINOL skeleton (1-6) were evaluated using toluene as the solvent at room temperature for 24 h (Table 1, entries 1-6), which all afforded good to excellent yields, but only with 27% ee being the best enantioselectivity (entry 6). After that, several 8H-BINOL-derived chiral phosphoric acids 7-9 were tested to be the reaction catalysts, among which 9-anthracenyl substituted phosphoric acid 7 gave the title product in 90% yield with 44% ee (entry 7). After that, the spirocyclic SPINOL-phosphoric acids were also evaluated as the catalysts on the enantioselective phosphinylation, but without getting further improvement in enantiocontrol (entries 10–12). Then, a series of solvents were screened in order to improve the enantioselectivity using 7 as the catalyst. When benzene was employed as the solvent instead of toluene, ee was sharply increased to 63%, albeit with only 81% yield (entry 13). Chloro-containing solvents such as CCl₄ and dichloromethane didn't show positive effect on this reaction, affording the desired product with only 2% and 16% ee, respectively (entries 14-15). Aprotic solvents such as EtOAc, MeCN, acetone, ether, 1,4-dioxane and THF all gave inferior enantiocontrol results (entries 16–21). To our surprise, methyl tert-butyl ether (MTBE) when being used as the solvent provided the best enantioselectivity (entry 22, 79% ee). Then, four different types of molecular sieves were introduced as the additive to the catalytic system (entries 23–26), among which 4 Å MS showed the best efficiency, providing 99% yield and 91% ee (entry 24). Finally, additional efforts at reaction optimization (temperature, catalyst loading, concentration, and activating reagents,) provided the similar levels of selectivity (for details, see Supplementary Tables 1-3). It is noted that the same level of enantioselectivity was observed by using benzene as solvent instead of MTBE (entry 27).

After the optimal conditions was set down, we then turned our attention to exploring the generality of this catalytic reaction (see Supplementary Methods). 3,4-Dihydroisoquinoline bearing substituents at different sites on the phenyl ring was firstly investigated. The results indicated that 5-substituted substrates bearing whatever electron-withdrawing or electron-donating groups were all readily phosphorylated with good to excellent yields and enantioselectivities (Fig. 2, 4baa-4faa). Sensitive functional groups like CN, Cl, Br and I (1a-e) were quite well tolerated under the catalytic conditions and above 93% ee was provided. Then, different types of functional groups at C-6 position of 3,4-dihydroisoquinoline were also studied. Electron-withdrawing groups (CN, F, Cl, Br) were all showed no negative effect on this asymmetric phosphinylation, providing the desired α-amino phosphine oxide in excellent yields and enantioselectivities (Fig. 2, 4gaa-4iaa). Substrates bearing methyl and iso-propyl

**Previous work:**

**a. Enantioselective phosphinylation of imines using diarylphosphine oxide**

**b. Enantioselective phosphinylation of imines using using dialkyl phosphite**

c. **This work:**
**P**hosphinylation of dihydroisoquinoline using diarylphosphine oxide

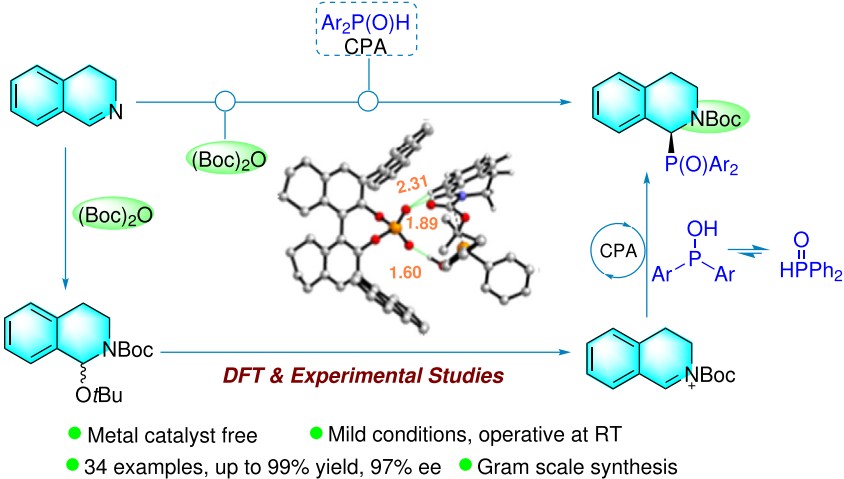

- Metal catalyst free
- Mild conditions, operative at RT
- 34 examples, up to 99% yield, 97% ee
- Gram scale synthesis

**Fig. 1 Works on phosphinylation of imines and 3,4-dihydroisoquinolines. a**, **b** Previous works on enantioselective phosphinylation of imines. **c** This work on phosphinlylation of dihydroisoquinoline using diarylphosphine oxide.

groups at C-6 position both showed inferior reactivity and enantioselectivity (Fig. 2, **4kaa** and **4laa**). After that, substituents at C-7 position were also investigated, among which phosphinylation of substrates **1m**, **1n**, **1o**, **1p** and **1r** all readily took place under the optimal conditions, offering the corresponding α-amino phosphine oxide products with good to excellent yields (76~92%) and excellent enantioselectivities (90~95%, ee), except **4qaa** (56% yield and 77% ee). What's noteworthy, substituents at C-8 position seemingly made a huge resistance for the title catalytic phosphinylation. Reaction of substrate **1s** was sluggish, giving the desired product (**4saa**) in 53% yield with only 3% ee. Moreover, no conversion to the desired product occurred with **1t** as substrate under the optimal conditions. These results might provide clues for understanding the catalytic cycle and even the enantiocontrol mechanism.

Then, the scope of nucleophilic secondary phosphine oxides was also studied. Firstly, the effects of substituents electron-property of substituents on the phenyl ring of secondary phosphine oxide were investigated, of which bis(4-(trifluoromethyl)phenyl)phosphine oxide (**2b**) and bis(4-(*tert*-butyl)phenyl)phosphine oxide (**2e**) both reacted readily with 6-Cl-3,4-dihydroisoquinoline (**1i**) with 86% ee and 93% ee respectively (Fig. 3, **4iba** and **4iea**). When 3,4-dihydroisoquinoline was introduced as the substrate instead of 6-Cl-3,4-dihydroisoquinoline, the conversion and enantioselectivity

were both slightly lower (Fig. 3, **4aba** and **4aea**). After that, di([1,1'-biphenyl]-4-yl)phosphine oxide (**2c**) and di-p-tolylphosphine oxide (**2d**) were also subjected to standard catalytic conditions reacting with 6-Cl-3,4-dihydroisoquinoline (**1i**) and 3,4-dihydroisoquinoline (**1a**) to provide the title products all with excellent enantiocontrol as listed in Fig. 3 (**4ica** and **4ida**; **4aca** and **4ada**). However, the strong electron-donating methoxy group (OMe) markedly decreased yield (40%) while maintaining good enantioselectivity (86% ee). What's noteworthy, when we changed the solvent into benzene, the yield significantly increased from 40% to 80% with the same enantioselectivity (83% ee) (Fig. 3, **4ifa**). The bis(naphthalen-1-yl)phosphine oxide (**2f**) and bis(naphthalen-2-yl) phosphine oxide (**2g**) were both proper nucleophiles to realize the phosphinylation of 6-Cl-3,4-dihydroisoquinoline with 91% ee and 96% ee respectively (Fig. 3, **4ifa** and **4iga**). Then, the steric effect of ortho-substituents was tested, product 4iia could be readily obtained with 88% yield and 90% ee when bis(2-methylphenyl) phosphine oxide (**2j**) was selected as the substrate, while methoxyl group played a negative role on this reaction, providing the desired product only with 35% yield and 86% ee (Fig. 3, **4ija**). Fortunately, the yield could also be increased to 78% by using benzene as the solvent. Product 3ika and 3ila were both smoothly yielded with 94% ee and 96% ee, respectively when steric hindered nucleophiles bis(3,5-dimethylphenyl)phosphine oxide (**2k**) and bis(3,5-di-*t*Bu-phenyl)phosphine (**2l**) oxide were employed under the standard

**Table 1 Catalytic conditions optimization[a].**

1, R = Ph, X = OH; 7, R = 9-anthracenyl, X = OH; **10**, R = 9-anthracenyl, X = Me;
**2**, R = adamantane, X = OH; **8**, R = 9-phenanthryl, X = OH; **11**, R = 9-anthracenyl, X = H;
**3**, R = 9-anthracenyl, X = OH; **9**, R = 9-phenanthryl, X = H; **12**, R = SiPh₃, X = H;
**4**, R = 9-phenanthryl, X = OH;
**5**, R = 9-anthracenyl, X = NTf;
**6**, R = 9-phenanthryl, X = NTf;

| Entry | CPA | Solvent | Yield (%)[b] | ee(%)[c] |
|---|---|---|---|---|
| 1 | **1** | toluene | 99 | 7 |
| 2 | **2** | toluene | 86 | 1 |
| 3 | **3** | toluene | 99 | 16 |
| 4 | **4** | toluene | 65 | 22 |
| 5 | **5** | toluene | 89 | 19 |
| 6 | **6** | toluene | 95 | 27 |
| 7 | **7** | toluene | 90 | 44 |
| 8 | **8** | toluene | 99 | 34 |
| 9 | **9** | toluene | 63 | 15 |
| 10 | **10** | toluene | 84 | 22 |
| 11 | **11** | toluene | 95 | 0 |
| 12 | **12** | toluene | 98 | 0 |
| 13 | **7** | benzene | 81 | 63 |
| 14 | **7** | CCl₄ | 99 | 2 |
| 15 | **7** | CH₂Cl₂ | 92 | 16 |
| 16 | **7** | EtOAc | 99 | 21 |
| 17 | **7** | MeCN | 99 | 2 |
| 18 | **7** | acetone | 22 | 16 |
| 19 | **7** | Ethyl ether | 35 | 14 |
| 20 | **7** | 1,4-dioxane | 57 | 12 |
| 21 | **7** | THF | 95 | 23 |
| 22 | **7** | MTBE | 96 | 79 |
| 23d | **7** | MTBE | 99 | 90 |
| 24e | **7** | MTBE | 99 | 91 |
| 25f | **7** | MTBE | 99 | 85 |
| 26g | **7** | MTBE | 99 | 88 |
| 27e | **7** | benzene | 99 | 90 |

[a]Reaction conditions: **1a** (0.2 mmol), Boc₂O (0.3 mmol) and CPA (5 mol%) at 50 °C for 0.5 h. Solvent (2 mL) and **2a** was added, and the mixture was stirred at rt for 24 h.
[b]Yield was determined by HPLC analysis.
[c]Determined by HPLC (Chiralcel OD-RH).
[d]with 3 Å MS.
[e]with 4 Å MS.
[f]with 5 Å MS.
[g]with 13X MS.

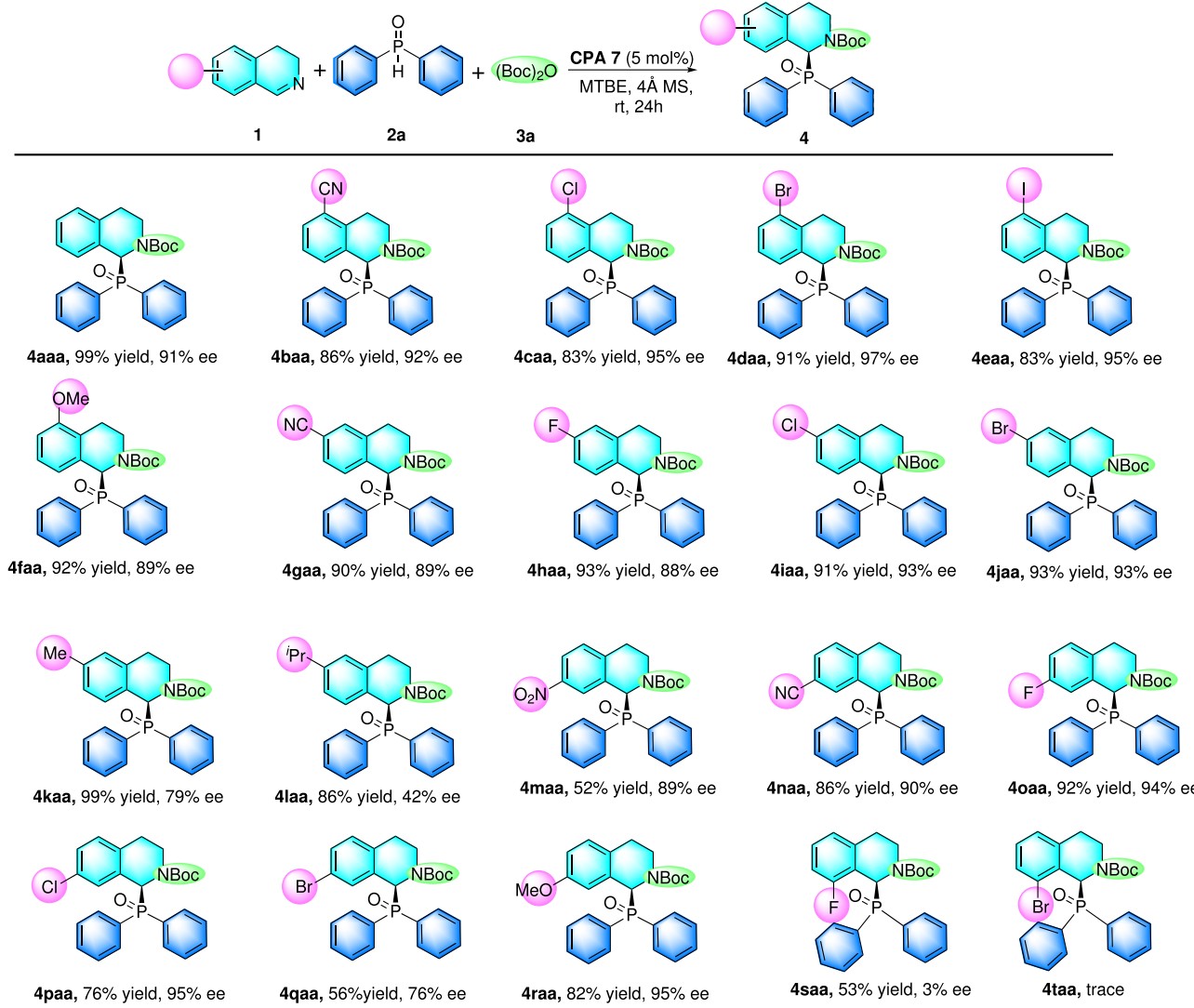

**Fig. 2 Scope of 3,4-Dihydroisoquinolines.** [a]Reaction conditions: **1** (0.2 mmol), Boc$_2$O **3a** (0.3 mmol) and **CPA 7** (5 mol%) at 50 °C for 0.5 h. 4 Å MS (50 mg), MTBE (2 mL) and **2a** (0.24 mmol) was added, and the mixture was stirred at room temperature for 24 h. Isolated yields were given here and ee value was determined by HPLC.

conditions. Unfortunately, more sterically hindered bis(2,4,6-trimethylphenyl)phosphine oxide (**2m**) turned out to be unreactive under such catalytic conditions. This result might be attributed to the steric hindrance effect of the double *ortho*-substituents.

To further demonstrate the practicability of our protocol, the enantioselective phosphinylation was carried out on a gram scale, offering the product **4aaa** with 95% yield and 91% ee under the optimal conditions (Fig. 4). After one recrystallization, the ee value increased to 99%. Under TFA/DCM or TMSCl/MeOH conditions, the Boc group was easily removed to produce **5aaa** in nearly quantitative yield with slight erosion of enantioselectivity. However, **5aaa** is unstable and easily racemizes in solution. Fortunately, racemization of Zinc salt **5aaa'** (solution) could be completely inhibited *via* removing the Boc group of **4aaa** with ZnBr$_2$, followed by condensation with iso(thio)cyanate compounds, providing divergent access to a wide spectrum of structurally diverse chiral THIQ derivatives that bear phosphine oxide and (thio)carbamide moieties (Fig. 4b). Among that, **6aaa** was obtained in 85% yield and 98% ee *via* condensation of freshly prepared **5aaa'** solution with isothiocyanatobenzene. After that, isocyanatoadmantane was also added into the **5aaa'** solution, providing **6aab** in 88% yield with 99% ee. Note that, enantiopure

1-isothiocyanatoethylbenzene and 1-isocyanatoethylbenzene were both smoothly transformed into **6aac** and **6aad** containing chiral scaffolds containing both phosphine oxide and (thio)carbamide moieties. Quinine-derived isothiocyanate was also verified to be effective for reaction with **5aaa'**, yielding product **6aae** in 86% yield and 99% ee. These chiral THIQ derivatives possibly functioned as chiral ligands or organocatalysts with potential applications in the field of asymmetric organocatalysis and Lewis base catalysis.

To provide insight into the catalytic mechanism of this reaction, several control experiments and NMR tracking experiments were carried out (Fig. 5, see Supplementary Figs. 1–7 for details). Firstly, treatment of 2,3-dihydroisoquinoline with Boc$_2$O for 30 min at 50 °C formed a new species, whose structure got verified to be tert-butyl 1-(tert-butoxy)-3,4-dihydroisoquinoline-2(1H)-carboxylate (**1a'**) by its [1]H NMR[47,48]. After that, compound **1a'** was subjected to the standard catalytic conditions in C$_6$D$_6$ as the solvent, of which the title product **4aaa** was formed in 99% yield with 90% ee, which indicates the possible intermediacy of compound **1a'** in the catalytic cycle (Fig. 5a). Then, substrate **1a** was treated with diphenyl phosphine oxide under the catalysis of CPA 7 in C$_6$D$_6$ at room temperature. After 12 h, the formation of

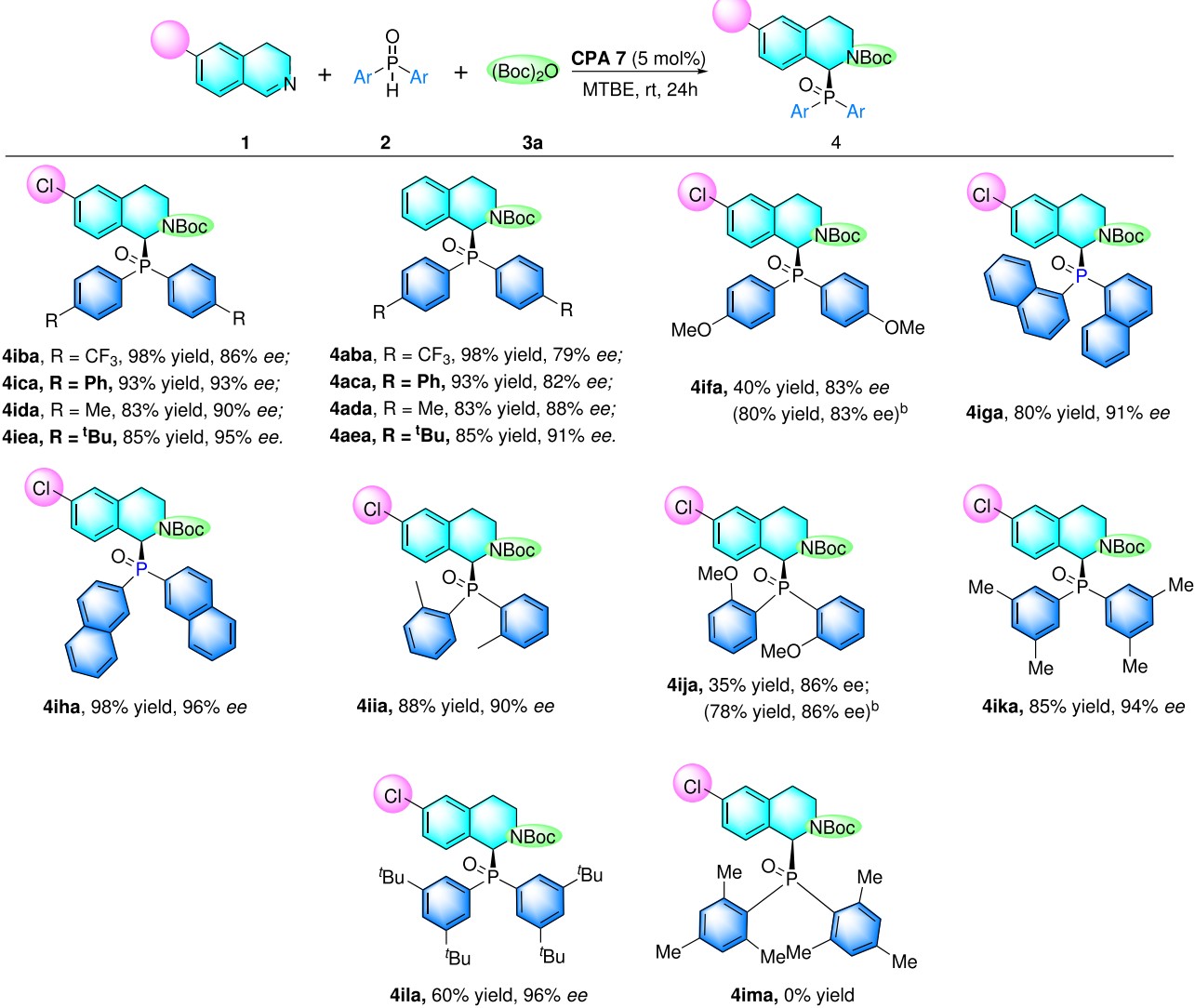

**Fig. 3 Scope of secondary phosphine oxides.** [a]Reaction conditions: **1** (0.2 mmol), Boc₂O **3a** (0.3 mmol) and CPA **7** (5 mol%) at 50 °C for 0.5 h. 4 Å MS (50 mg), MTBE (2 mL) and **2a** (0.24 mmol) was added, and the mixture was stirred at room temperature for 24 h. Isolated yields were given here and ee value was determined by HPLC (Chiralcel OD-RH). [b]Benzene was used instead of MTBE.

diphenyl(1,2,3,4-tetrahydroisoquinolin-1-yl)phosphine oxide (**1b'**) was observed. Then, treatment of isoquinoline (**1b'**) with Boc₂O in the presence of CPA **7** formed racemic title product **4aaa** in 99% yield (Fig. 5b). These results show that the asymmetric reaction process does not go through via the intermediate **1b'**. Finally, the experiment that **1a** directly reacted with diphenyl phosphine oxide in the presence of Boc₂O under the catalysis of phosphoric acid was carried out, of which the desired product **4aaa** was offered in 99% yield along with only 40% ee (Fig. 5c). The result indicated that the phosphinylation reaction occurred in a stepwise manner and the clean formation of **1a'** played pivotal role in the step of enantiocontrol.

DFT calculations were also conducted to understand the mechanism and the origin of the enantioselectivity of this reaction. The catalyst CPA **7** and substrate **1a** were considered in our DFT calculations. On the basis of the mechanistic discussion above, we calculated the energy profile for the pathway shown in Fig. 5a. As shown in Fig. 6, the mechanistic cycle consists of three major stages. Stage 1 (**1a** → **1a-1**) is related to the electrophilic addition of (Boc)₂O to **1a** with an energy barrier of 16.6 kcal/mol. In stage 2, decarboxylation of **1a-1** to give **1a'** is rate-determining for the whole pathway, with an overall activation free-energy

barrier of 30.4 kcal/mol (the energy of **TS3** related to **1a'**). Stage 3 (**1a'** → **4aaa**) corresponds to the enantioselective phosphinylation of **1a'** with a stepwise nucleophilic substitution mechanism via an isoquinolinium intermediate involved. In stage 3, the bonding of diphenylphosphine oxide with CPA **7** increases the acidity of CPA **7**. The diphenylphosphine-bonded CPA **7** then protonates **1a'** and releases a molecular or tBuOH to give the isoquinolinium **1a'-1**. The phosphinylation of the isoquinolinium is enantio-determining step. This catalytic mode is consistent with previous closely related work[49–51]. The energy difference between the enantiodetermining transition states **TS-5(S)** and **TS-5(R)** was calculated to be 2.9 kcal/mol (98% ee), which correctly reproduces the enantiomeric preference observed experimentally (91% ee for **S-4aaa**).

By carefully checking the structures of **TS5(R)** and **TS5(S)** (Fig. 7), we found that the CPA **7**-phosphonate has stronger hydrogen bonding interaction with isoquinolinium in the favorable transition state **TS5(S)** than that in the **TS5(R)**. The hydrogen at C-8 position of isoquinoline is involved in the hydrogen bonding interactions with the distance of C8-H and CPA **7**-phosphonate in **TS5(S)** at 2.31 Å which is shorter than that in **TS5(R)** (2.45 Å). This finding suggests that the aryl

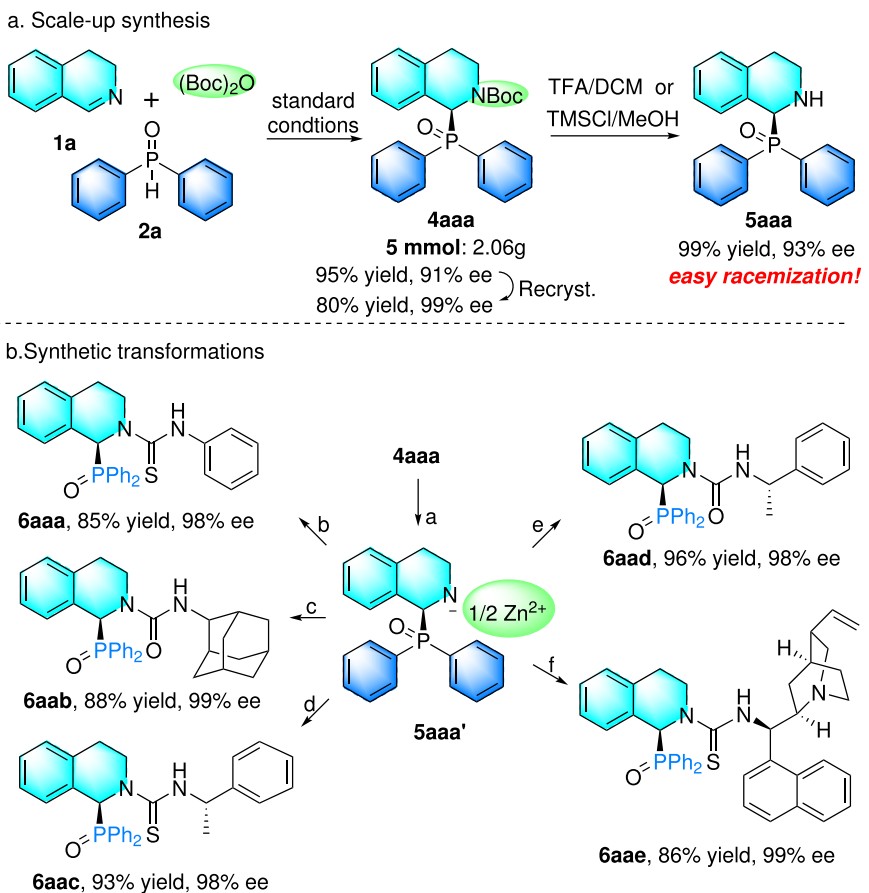

**Fig. 4 Scale-up synthesis and synthetic transformations. a** Scale-up synthesis, [a]Reaction conditions: **4aa** (1 mmol), ZnBr₂ (1.2 mmol) in DCM (10 mL) at room temperature for 12 h, see Supplementary Notes. No further post-processing required; **b** Synthetic transformations, b-fRelated iso(thio)cyanate compounds (1.2 mmol) were added, and the mixture was stirred at room temperature for 24 h. Isolated yields were given here and ee value was determined by HPLC.

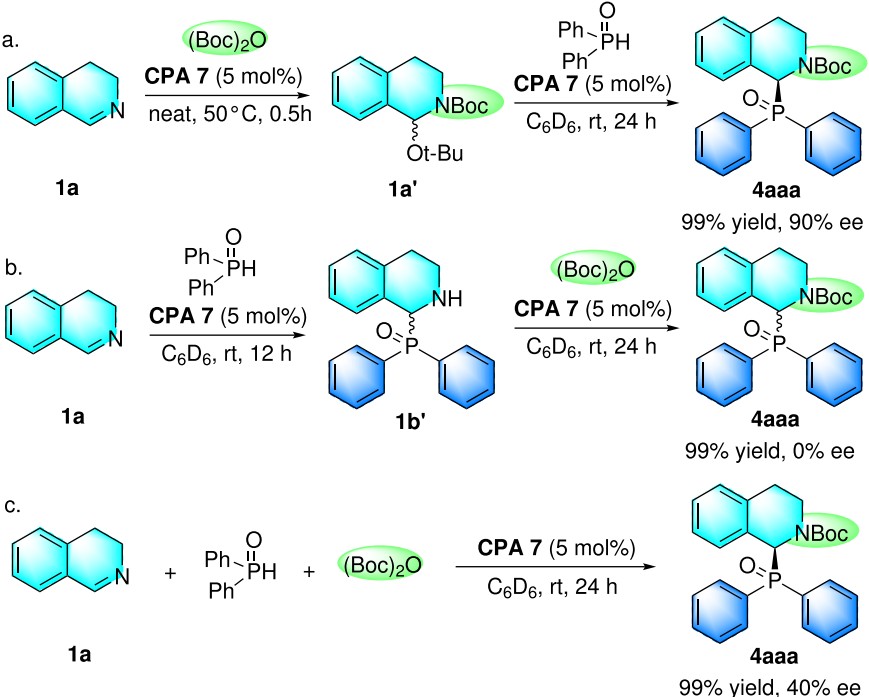

**Fig. 5 Control and NMR tracking experiments. a** Control experiment to verify the intermediate **1a'**; **b** Control experiment to exclude the intermediate **1b'**; **c** Control experiment to show the reaction running in a stepwise manner.

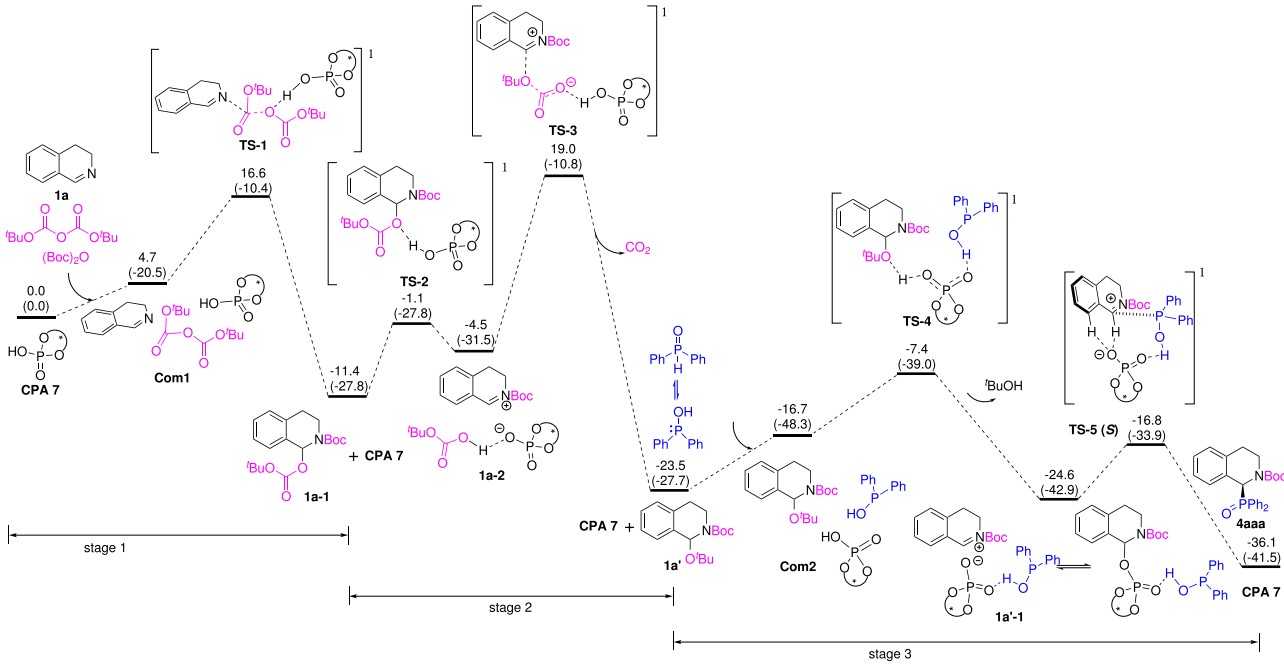

**Fig. 6 Energy profile calculated for the phosphoric acid catalyzed enantioselective phosphinylation of enantioselective phosphinylation of isoquinoline 1a.** Relative free energies and electronic energies (in parentheses) are given in kcal/mol.

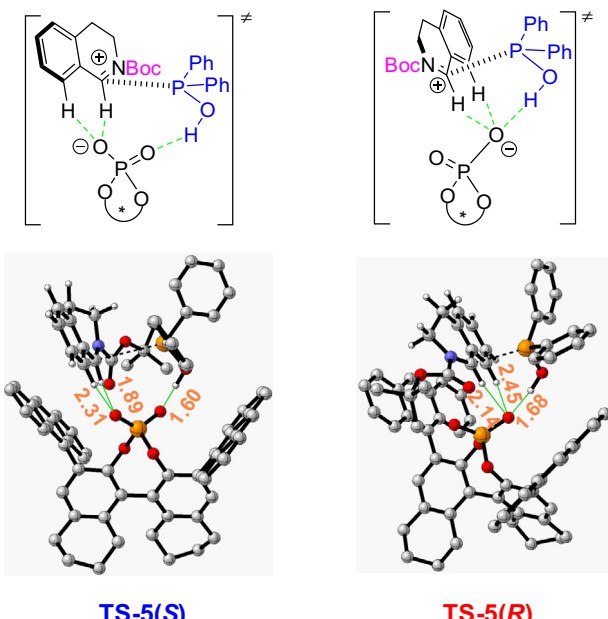

**Fig. 7 Structures of transition states of phosphinylation of the isoquinolinium.** Bond distance is given in Å. Unimportant H atoms are omitted for clarity.

hydrogen at C-8 position also plays a role in the determination of the stereoselectivity of this reaction, consistent with the experimental observations shown in Fig. 2 that substituents at C-8 position (**4saa**) give poor enantioselectivity.

## Conclusions
In summary, we have developed an efficient and mild methodology for the asymmetric phosphinylation of 3,4-dihydroisoquinoline (up to 99% yield and 97% ee) using secondary phosphine oxide for

the first time. Applicability of this protocol was well demonstrated *via* gram-scale synthesis and preparing several potential chiral organocatalysts or ligands through the derivatization of the products. Experimental studies and DFT calculations support the intermediacy of *tert*-butyl 1-(*tert*-butoxy)-3,4-dihydroisoquinoline-2(1*H*)-carboxylate. Also, experimental studies and DFT calculations suggest that hydrogen bonding strength between the phosphonate and isoquinolinium plays pivotal role in the determination of the stereoselectivity in this reaction.

## Methods
**General procedure for CPA-catalyzed Phosphinylation of 3,4-Dihydroisoquinolines.** A mixture of 3,4-dihydroisoquinoline (0.2 mmol), CPA catalyst (5 mol%) and (Boc)₂O (0.3 mmol) was stirred at 50 °C for 0.5 h. Then 4 Å MS (50 mg), MTBE or benzene (2 mL) and diarylphosphine oxide (0.24 mmol) was added, and the reaction was stirred at room temperature for 24 h. The reaction mixture was concentrated under reduced pressure. The residue was purified by flash column chromatography with PE/EA (2/1) to obtain *tert*-butyl 1-(diphenylphosphoryl)-3,4- dihydroisoquinoline-2(1*H*)-carboxylates.

## Data availability
All data generated during this study are included in this article and Supplementary Information. Experimental procedure, conditions optimization, control and tracking experiments, product characterization and DFT calculations are provided in the Supplementary Information. The NMR spectra of all compounds are available in Supplementary Data 1. HPLC chromatograms of the chiral products are available in Supplementary Data 2. The Cartesian coordination of the key structures are provided in Supplementary Data 3. It can be declared that all the relevant data are provided in the article and its Supplementary Information files.

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

## Acknowledgements

Y-N.D., acknowledges the financial support from Chemistry and Chemical Engineering Guangdong Laboratory (Grant nos. 2011006 and 2132013) and the Special Fund for the Sci-tech Innovation Strategy of Guangdong Province (no. 210730166882026).

## Author contributions

Y.G., Y-N.D., and J.X. conceived and directed the project. Y.G., Z.G., N.L., and J.L. performed the experiments. Y-N.D. performed the theoretical calculations. X.B., and Z.G. analyzed the results. Y.G. and J.X. wrote the manuscript.

## Competing interests

The authors declare no competing interests.
