## [Peer Review File · Communications Chemistry]

Reviewers' comments:

Reviewer #1 (Remarks to the Author):

Chiral phosphorous-containing compounds constitute an important class of chiral ligands and organocatalysts for asymmetric reactions, and are common building blocks for a number of bioactive natural products and marketed drugs. The manuscript describes a chiral phosphoric acid-catalyzed enantioselective phosphinylation of 3,4-dihydroisoquinolines with diarylphosphine oxides providing a family of chiral α -amino diarylphosphine oxide. Applicability of this protocol was well demonstrated via gram-scale synthesis and preparing several potential chiral organocatalysts or ligands. Control and NMR tracking experiments, and computational studies elucidated the determination of the stereoselectivity in this reaction. The work can be of utility to researchers interested in the organic chemistry and medicinal chemistry. Thus I recommend the publication of this manuscript after following the below modification is made.

1. The authors in the manuscript are not consistent with the SI.
2. Some products should be purified to get good quality NMR, such as compound 4gaa
3. The chiral phosphoric acid catalyst is the key success for the asymmetric new reaction, thus some important references about the development of chiral phosphoric acid should be cited, such as *Angew. Chem. Int. Ed.* 2004, 43, 1566; *J. Am. Chem. Soc.* 2004, 126, 5356 for BINOL-CPAs, *J. Org. Chem.* 2010, 75, 8677; *Angew. Chem. Int. Ed.* 2019, 44, 15824 for SPINOL-CPAs; reviews: *Chem. Rev.* 2007, 107, 5744; *Chem. Rev.* 2014, 114, 9047; *Chinese Journal of Chemistry*, DOI: 10.1002/cjoc.202000446.

Reviewer #2 (Remarks to the Author):

The manuscript by Guo, Gao, Duan, and coworker reports a chiral phosphoric acid (CPA)-assisted asymmetric phosphinylation of 3,4-dihydroisoquinolines using secondary phosphine oxides to construct phosphinylated tetrahydroisoquinolines (THIQ). Mechanistically, 3,4-dihydroisoquinoline reacts with di-tert-butyl dicarbonate (Boc₂O) in the presence of CPA to form a tert-butyl 1-(tert-butoxy)-3,4-dihydroisoquinoline-2(1H)-carboxylate intermediate. CPA-mediated formation of isoquinolinium intermediate then sets the stage for the enantioselective phosphinylation to afford the target optically active α -amino diarylphosphine oxides.

The combination of the substructures α -amino diarylphosphine oxides and THIQs is an interesting notion, and the paper is overall technically sound. The authors concluded the first use of secondary phosphine oxides in the asymmetric phosphinylation of 3,4-dihydroisoquinolines (DHIQs) through an efficient and mild protocol. However, the entire study itself does not offer enough novelty to warrant publication in *Communications Chemistry*. It should be noted that the same authors, Gao and Guo, recently published a paper featuring the CPA-catalyzed enantioselective phosphonation of THIQs and DHIQs [*Chem. Commun.* 2022, 58, 9393–9396]. The mentioned paper discusses a strikingly similar strategy (CPA-mediated Reissert-type reaction), and the only difference is the phosphorus nucleophiles (phosphonates versus phosphine oxides). Curiously, this manuscript has failed to mention the previous work.

Moreover, the chemistry and protocol provided by the authors is in fact excellent but just did not offer new conceptual insights, especially when considering the familiar approach that they had utilized before. Further development of the method should be anchored on novel phosphinylation approaches (perhaps with a fresh perspective on catalysis), and not just on tweaking the substrates or nucleophiles, which would seem like a mere extension of the previous protocol/s reported.

This author would like to recommend that this manuscript be published in a specialized/discipline-specific journal instead.

Additional comments:

1. Three of the listed authors in the Supplementary Information were absent in the main manuscript and thus this inconsistency should be addressed.

Reviewer #3 (Remarks to the Author):

The authors reported enantioselective phosphinylation of 3,4-dihydroisoquinolines catalyzed by chiral phosphoric acid. The secondary phosphine oxides are firstly used as nucleophiles for phosphinylation of 3,4-dihydroisoquinolines. They provide control experiments to investigate the reaction mechanism via the key intermediate of tert-butyl 1-(tert-butoxy)-3,4-dihydroiso-quinoline-2(1H)-carboxylate, followed by C-P bond formation. However, the computational results and analysis are too rough to clarify the origin of stereo-selectivity. I cannot recommend acceptance of the current version of manuscript.

Major point:

1. The key transition states and intermediates involve obvious hydrogen bond interactions, but the functions used for structural optimization does not add dispersion corrections. I doubt that this kind of computational studies may lead to qualitative error, and the mechanistic result may not be suspected.

2. The complex adducts before TS-1 and TS-4 should be shown in the energy profile, because the electronic energy change (ΔE^\ddagger) from zero point to TS-1 is negative, as opposed to the free energy change (ΔG^\ddagger). Then the relative electron energy of the complex adducts must be lower than zero point, and its relative free energy is likely to be lower than zero point. So using current zero point to TS-1 may underestimate the reaction barrier.

3. It can be seen from the control experiments (scheme 5b) that the Boc- group on N is very important for improving stereoselectivity, and it can be roughly seen that the tert-butyl group seems to contribute to the stability differences between the enantiomers of TS5. The authors should provide a reasonable explanation either from control experiments or computational studies. And further understanding the origin of stereoselectivity is also preferred.

Minor points

4. There are two errors in Figure 1. One is the Chemdraw structure of TS-2 was incorrectly drawn where "NBoc" was drawn instead of "NH". And the other is TS-5(S) is written as TS-4(S).

5. The Chemdraw diagrams and the corresponding atomic number should be provided for illustrating enantioselectivity in Figure 2. The quality and resolution of Scheme 1c and Figure 2 are hard to see what the authors are trying to show. Please, improve the quality of the Figure.

6. The Cartesian coordination of the key structures studied are not given, and the specific structure cannot be accessible.

Point-by-Point Response

Reviewer #1 (Remarks to the Author):

Chiral phosphorous-containing compounds constitute an important class of chiral ligands and organocatalysts for asymmetric reactions, and are common building blocks for a number of bioactive natural products and marketed drugs. The manuscript describes a chiral phosphoric acid-catalyzed enantioselective phosphinylation of 3,4-dihydroisoquinolines with diarylphosphine oxides providing a family of chiral α -amino diarylphosphine oxide. Applicability of this protocol was well demonstrated via gram-scale synthesis and preparing several potential chiral organocatalysts or ligands. Control and NMR tracking experiments, and computational studies elucidated the determination of the stereoselectivity in this reaction. The work can be of utility to researchers interested in the organic chemistry and medicinal chemistry. Thus I recommend the publication of this manuscript after following the below modification is made.

1. The authors in the manuscript are not consistent with the SI.

Response: *We have made the correction and kept the authors in the manuscript consistent with that of Supplementary Information.*

2. Some products should be purified to get good quality NMR, such as compound 4gaa

Response: *As suggested by this reviewer, we have further purified the products including product 4gaa and the better NMR spectrums were added into the revised Supplementary Information.*

3. The chiral phosphoric acid catalyst is the key success for the asymmetric new reaction, thus some important references about the development of chiral phosphoric acid should be cited, such as Angew. Chem. Int. Ed. 2004, 43, 1566; J. Am. Chem. Soc. 2004, 126, 5356 for BINOL-CPAs, J. Org. Chem. 2010, 75, 8677; Angew. Chem. Int. Ed. 2019, 44, 15824 for SPINOL-CPAs; reviews: Chem. Rev. 2007, 107, 5744; Chem. Rev. 2014, 114, 9047; Chinese Journal of Chemistry, DOI: 10.1002/cjoc.202000446.

Response: *As suggested by this referee, we have added the relevant references in the modified manuscript which was cited as ref. 12a-12g which was highlighted in bright yellow color.*

Reviewer #2 (Remarks to the Author):

The manuscript by Guo, Gao, Duan, and coworker reports a chiral phosphoric acid (CPA)-assisted asymmetric phosphinylation of 3,4-dihydroisoquinolines using secondary phosphine oxides to construct phosphinylated tetrahydroisoquinolines (THIQ). Mechanistically, 3,4-dihydroisoquinoline reacts with di-tert-butyl dicarbonate (Boc₂O) in the presence of CPA to form a tert-butyl 1-(tert-butoxy)-3,4-dihydroisoquinoline-2(1H)-carboxylate intermediate.

CPA-mediated formation of isoquinolinium intermediate then sets the stage for the enantioselective phosphinylation to afford the target optically active α -amino diarylphosphine oxides.

The combination of the substructures α -amino diarylphosphine oxides and THIQs is an interesting notion, and the paper is overall technically sound. The authors concluded the first use of secondary phosphine oxides in the asymmetric phosphinylation of 3,4-dihydroisoquinolines (DHIQs) through an efficient and mild protocol. However, the entire study itself does not offer enough novelty to warrant publication in Communications Chemistry. It should be noted that the same authors, Gao and Guo, recently published a paper featuring the CPA-catalyzed enantioselective phosphonation of THIQs and DHIQs [Chem. Commun. 2022, 58, 9393–9396]. The mentioned paper discusses a strikingly similar strategy (CPA-mediated Reissert-type reaction), and the only difference is the phosphorus nucleophiles (phosphonates versus phosphine oxides). Curiously, this manuscript has failed to mention the previous work.

Moreover, the chemistry and protocol provided by the authors is in fact excellent but just did not offer new conceptual insights, especially when considering the familiar approach that they had utilized before. Further development of the method should be anchored on novel phosphinylation approaches (perhaps with a fresh perspective on catalysis), and not just on tweaking the substrates or nucleophiles, which would seem like a mere extension of the previous protocol/s reported.

This author would like to recommend that this manuscript be published in a specialized/discipline-specific journal instead.

Response: *Thanks for the suggestions of this reviewer. Firstly, we have to mention that this manuscript and the above-mentioned work (Chem. Commun. 2022, 58, 9393–9396, featuring in the CPA-catalyzed enantioselective phosphonation of isoquinolines) were nearly submitted at the same time to two different journals, respectively. And the submission and review process of this manuscript was severely prolonged owing to some reasons maybe including the influence of COVID-19. What's more, the work described in this manuscript showed obvious difference as compared with that reported in our previous work. It represents the first report for enantioselective phosphonation of 3,4-dihydroisoquinoline using diarylphosphine oxides, which was quite challengeable owing to the facile aromatization of the substrate. It was also observed in our previous work (Chem. Commun. 2022, 58, 9393–9396) on the CPA-catalyzed enantioselective phosphonation of isoquinolines, which was found to have obvious limitations owing to the difficulties lying in the further transformation of the products led by its facile racemization via thermodynamically favored aromatization process.*

Additional comments:

1. Three of the listed authors in the Supplementary Information were absent in the main manuscript and thus this inconsistency should be addressed.

Response: *We have made the correction and kept the authors in the manuscript consistent with that of Supplementary Information.*

Reviewer #3 (Remarks to the Author):

The authors reported enantioselective phosphinylation of 3,4-dihydroisoquinolines catalyzed by chiral phosphoric acid. The secondary phosphine oxides are firstly used as nucleophiles for phosphinylation of 3,4-dihydroisoquinolines. They provide control experiments to investigate the reaction mechanism via the key intermediate of tert-butyl 1-(tert-butoxy)-3,4-dihydroiso-quinoline-2(1H)-carboxylate, followed by C-P bond formation. However, the computational results and analysis are too rough to clarify the origin of stereo-selectivity. I cannot recommend acceptance of the current version of manuscript.

Response: Thanks for the reviewer's insightful summary and general comments.

Major point:

1. The key transition states and intermediates involve obvious hydrogen bond interactions, but the functions used for structural optimization does not add dispersion corrections. I doubt that this kind of computational studies may lead to qualitative error, and the mechanistic result may not be suspected.

Response: We thank the reviewer for the insightful comments. The method used in our calculation referred to the publications by the very experienced groups on calculational chemistry when calculating the phosphoric acid-catalyzed reactions (Kendall N. Houk et. al Science **2018**, 361, eaas8707 and Z. Lin et. al ChemCatChem **2020**, 12, 5053-5057). We believe the methods used in this study have been proven to be reliable for the calculations of the phosphoric acid-catalyzed reactions. To address the reviewer's concern, we re-optimized the enantio-determining transition states **TS5(S)** and **TS5(R)** at the B3LYP-D3/6-31G* level. As shown in the following table, the energy difference is consistent with that obtained from our original calculations. We have added this table in the SI.

	B3LYP	B3LYP-D3
$\Delta\Delta G^*$	1.9	1.7
ee (%)	91.6	89.5

2. The complex adducts before TS-1 and TS-4 should be shown in the energy profile, because the electronic energy change (ΔE^\ddagger) from zero point to TS-1 is negative, as opposed to the free energy change (ΔG^\ddagger). Then the relative electron energy of the complex adducts must be lower than zero point, and its relative free energy is likely to be lower than zero point. So using current zero point to TS-1 may underestimate the reaction barrier.

Response: The van der Waals complex before **TS-1** was assembled by the compounds **CPA 7**, **(Boc)₂O** and **1a**. This process is thermodynamically unfavorable with the free energy change (ΔG) at 4.7 kcal/mol and cannot change the relative free energy. It means that the van der Waals interaction cannot counteract the entropy cost in this three-to-one process. For the same reason, assembling the der Waals complex before **TS-4** is also unfavorable. We have added the van der Waals complexes in the revised manuscript.

3. It can be seen from the control experiments (scheme 5b) that the Boc- group on N is very important for improving stereoselectivity, and it can be roughly seen that the tert-butyl group seems to contribute to the stability differences between the enantiomers of TS5. The authors should provide a reasonable explanation either from control experiments or computational studies. And further understanding the origin of stereoselectivity is also preferred.

Response: The H-bonding interactions between the phosphoric acid (P-O and P=O) and the reactant fragment(s) must maintain in the enantio-determining transition states to effectively control the enantioselectivity (for example the interactions shown in red and blue in **TS-5(S)**). This basic understanding of the nature of chiral phosphoric acid catalysis guides us on how to design control experiments and calculations. The transition states in the pathway of **Scheme 5b** cannot simultaneously maintain the H-bonding interaction both on the P-O and P=O parts.

TS-5 (S)

Minor points

4. There are two errors in Figure 1. One is the Chemdraw structure of TS-2 was incorrectly drawn where “NBoc” was drawn instead of “NH”. And the other is TS-5(S) is written as TS-4(S).

Response: Thanks for this reviewer’s suggestion and we have revised the typos, which was highlighted in bright yellow color in the revised manuscript.

5. The Chemdraw diagrams and the corresponding atomic number should be provided for

illustrating enantioselectivity in Figure 2. The quality and resolution of Scheme 1c and Figure 2 are hard to see what the authors are trying to show. Please, improve the quality of the Figure.

Response: *Thanks for this reviewer's suggestion and we have improved these two figures, which was highlighted in bright yellow color in the revised manuscript.*

6. The Cartesian coordination of the key structures studied are not given, and the specific structure cannot be accessible.

Response: *Thanks for this reviewer's suggestion and the Cartesian coordination has been given as a supporting material in the xyz form.*

REVIEWERS' COMMENTS:

Reviewer #1 (Remarks to the Author):

The authors have revised the paper, and give full modification.
I'm satisfied with the authors' response.
I recommend the publication of this manuscript.

Reviewer #3 (Remarks to the Author):

I think the revised paper had addressed all my concerns. I therefore support this paper to be published.